# Analysis of Predictive Factors for Successful Vascular Anastomoses in a Sheep Uterine Transplantation Model

**DOI:** 10.3390/jcm11185262

**Published:** 2022-09-06

**Authors:** Claire Le Gal, Marie Carbonnel, Vincent Balaya, Christophe Richard, Valerie Gelin, Laurent Galio, Olivier Sandra, Barbara Hersant, Romain Bosc, Johanna Charton, Pascale Chavatte-Palmer, François Vialard, Raphael Coscas, Jean-Marc Ayoubi

**Affiliations:** 1UVSQ, INRAE, BREED, Paris-Saclay University, 78350 Jouy-en-Josas, France; 2Department of Gynecology and Obstetrics, Foch Hospital, 92150 Suresnes, France; 3National Veterinary School, BREED, 94700 Maisons-Alfort, France; 4MIMA2 Platform, INRAE, 78350 Jouy-en-Josas, France; 5Department of Maxillofacial and Plastic & Reconstructive Surgery, Henri Mondor University Hospital, 94000 Creteil, France; 6Department of Vascular Surgery, Amboise Pare University Hospital, AP-HP, 92100 Boulogne-Billancourt, France; 7Department of Genetics, Medical Biology Laboratory, Poissy-St Germain en Laye University Hospital, 78300 Poissy, France; 8UMR 1018, Inserm-Paris11-CESP, Versailles Saint-Quentin-en-Yvelines University, 78000 Versailles, France

**Keywords:** uterus, transplantation, sheep, vascular, anastomoses

## Abstract

Uterine transplantation is becoming an increasingly realistic therapeutic for uterine infertility. Surgical training on large animal models such as sheep is a prerequisite for establishing a program in humans. The objective of our study was to analyze the predictive factors for successful vascular anastomoses. We performed 40 autotransplants that involved end-to-side anastomoses from the uterine to the external iliac vessels. We analyzed vessel results in terms of success or failure; a total of 78.7% of arterial and 82.9% of venous anastomoses were successful in the immediate postoperative period. In multivariate analysis, independent factors associated with immediate successful vein anastomoses were as follows: a short warm ischemia time (<2 h, OR = 0.05; 95% CI [0.003–0.88], *p* = 0.04), the absence of any anastomotic complications (OR = 0.06; 95% CI [0.003–0.099], *p* = 0.049), and their realization by a vascular surgeon (OR = 29.3; 95% CI [1.17–731.9], *p* = 0.04). Secondly, we showed that an increase in lactate levels greater than 2.72 mmol/L, six hours after reperfusion was predictive of failure, with a sensibility of 85.7% and a specificity of 75.0%. In order to perfect the management of vascular anastomoses by a vascular surgeon, training on animal models and in microsurgery are mandatory in establishing a uterine transplantation program in humans.

## 1. Introduction

Absolute uterine infertility affects around 0.2% women of childbearing age around the world [1,2]. It is mainly due to a congenital Mullerian malformation such as Mayer–Rokitansky–Küster–Hauser (MRKH) syndrome [3]. Until recently, these women could not become pregnant, and the only option to become a mother was through adoption or gestational surrogacy, which is prohibited in many countries including France [4]. Uterine transplantation (UTx) stands as an alternative and promising treatment, yet it still is currently in the experimental stage.

The first breakthrough was obtained by a Swedish team led by Pr Brännström, with the first live birth in 2014 after UTx from a live donor [5]. Currently, several international teams are working on this project, and progress in this area is rapid. The feasibility of UTx from mainly living but also deceased donors has been proven [6]. Currently, 80 UTx procedures have been performed worldwide, with more than 30 live births [7,8].

UTx is still at the experimental stage because of the complexities involved in the surgical procedure, including uterine vessel dissection and vascular anastomoses. Live donor surgery is a much more extensive procedure than a simple hysterectomy; in order to obtain a sufficient length of free bilateral uterine vascular pedicles, about 6 cm, required to achieve direct end-to-side anastomoses with the external iliac vessels [9,10]. The trickiest part is performing venous dissection, since the small uterine vein has plexuses and many anatomical variants without rigorous systematization [11]. Although most teams use the uterine veins as an outlet, some use the ovarian vein, utero-ovarian vein, or a combination of both, depending on the quality of the vessels (Figure 1) [12,13]. This stage is a source of major complications, including hemorrhages and ureteral damage. Then, the creation of four anastomoses (two veins and two arteries) on deeply located vessels of small caliber [14], complicates the surgery on the recipient. In the literature, 30% of the grafts had to be explanted, mainly due to vascular thromboses [6,15,16].

Based on experimentation that was carried out with various animal models, ewes represent the most appropriate model for surgical training in UTX due to their similarities with humans in terms of body size, pelvic anatomy, and uterine vessel size [17,18]. Thus, we established an experimental model of uterine autotransplantation in ewes, as part of a uterine transplantation project in women. The first uterine transplantation in France was performed by our team in 2019, which led to a live birth in 2021 [19].

The aim of our study was to identify potential factors for successful vascular anastomoses in a sheep model of uterine autotransplantation.

## 2. Materials and Methods

### 2.1. Ethical Approval

The study was approved by the INRAE-AgroParisTech Animal Ethics Committee (CEEA 45), and the French Ministry of National Education, Teaching and Research (APAFIS#7380-2016062417IS6424 v5, APAFIS#13118–2017122219387116_v1, and APAFIS#19628-201810041821402 v4). All animals were treated according to Directive 2010/63/EU of the European Parliament relative to the protection of animals, and compliant with the ARRIVE criteria [20].

### 2.2. The Team

The manipulations were performed on the INRAE site of Jouy-en-Josas at the CIMA structure, including an operating room for animal experimentation. Our team included surgeons of complementary specialties (gynecologist-obstetricians, vascular surgeons, and non-vascular surgeons), zootechnicians, and veterinarians who were specialized in animal reproduction research. The gynecologic surgeons had expertise in advanced benign and oncologic surgeries. Before beginning our program, team members had participated in three ovine autotransplants with the Swedish team (Goteborg, Sweden), and had performed two uterine dissections on ewe cadavers. The specialty of transplant surgeon does not exist in France; vascular anastomoses are performed by organ specialists, depending on the graft performed. Vascular anastomoses were initially performed by non-vascular surgeons who were experts in microscopic surgery, from the first case to the fifteenth, and then by a vascular surgeon who performed the anastomoses from the sixteenth to the fortieth case.

### 2.3. Animals and Experimental Protocol

We operated on adult ewes of the Prealpes and Romane breeds. The females were synchronized using a vaginal progesterone pessary that was inserted 10 days before surgery, as routinely performed in animal husbandry. The ewes were fasted 24 h before surgery.

Since 2017, forty ewes have been operated on using three different procedures. The first included five ewes, and this allowed us to establish a standardized operative protocol and confirm the feasibility of the surgery. A 3-hour reperfusion was performed before the various samples were taken and the ewe was euthanized. The second series included 11 ewes, and was specifically designed to compare the performance of one venous anastomosis versus two. Between groups, no differences were found in terms of molecular, cellular, and histologic findings [21].

For the third series the protocol was modified, and the ewes were awakened at the end of the procedure. In this series, in accordance with the 3R principles (Replacement, Reduction, and Refinement), only 15 ewes were initially required: 5 in each group (2 groups with different organ preservation fluids, and a control group). They were euthanized 4 days after surgery. Those ewes that were euthanized for intraoperative complications or that died as a result of another nonvascular cause were excluded from the short-term analyses. An additional 9 ewes were required. The experimental protocol is summarized in Figure 2.

### 2.4. Anatomy of the Ovine Model

The uterus of the sheep is bicornuate, being composed of two uterine horns with a very short uterine body terminated by the cervix (Figure 3). The uterus is vascularized by an uterine artery, and drained by one utero-ovarian vein without an ovarian vein [22,23]. As described in humans, many anatomic variations can be observed in sheep.

### 2.5. Anesthesia

The animals were brought into the operating room and cared for by two animal specialists. General anesthesia and monitoring throughout the procedure were performed according to a protocol that was detailed in our previous publication [17].

### 2.6. Surgical Technique

#### 2.6.1. Uterine Dissection

In order to achieve a compromise between depth of field and magnification, custom X2.5 magnifiers were adapted prior to this study (Orascopic HDL™ Endehavour™). A para-median incision was made from the pubis to the umbilicus, in order to avoid the large subcutaneous mammary vein. The rumen, colon, and intestine were wrapped in wet drapes, placed in the upper abdomen, and held in place with retractors (Bookwalter^®^III retractor system, Symmetry Surgical).

In order to facilitate dissection of the broad ligament, the right round ligament was ligated, cut, and oriented medially. The utero-ovarian vein was dissected to the internal iliac vein. Next, we tracted over the umbilical artery to assist in the dissection of the uterine artery to the internal iliac artery. We performed the dissection upstream of its posterior branch, which was ligated, in order to obtain a better vascular diameter from the eleventh ewe. In order to perform this dissection, a complete ureterolysis was performed up to the end of the bladder. The procedure was performed for both sides. The bladder was then detached from the uterus, and dissection was continued along the cervix until a 2-cm vaginal cuff was obtained. The vaginal vessels were ligated. The ovarian artery above the utero-ovarian vein was also ligated and resected. The uterus was removed with a large portion of the posterior peritoneum, utero-sacral ligaments, and posterior vaginal wall, in order to enhance later the fixation of the uterus and avoid torsion. Vascular clamps were placed first at the origin of the uterine arteries and utero-ovarian veins, and then all vessels were resected. The origins of the uterine arteries and utero-ovarian veins up to the internal iliac artery and vein were ligated. Heparin was injected intravenously before clamping the uterine vessels.

#### 2.6.2. Back Table

The uterus was immediately placed on ice at 4 °C and flushed with a heparinized ringer lactate solution at a concentration of 5000 IU/L of heparin, until a clear liquid was obtained at the level of the two venous returns. The remaining fatty tissue was removed distal to the arteries and veins, in order to facilitate anastomoses (Figure 4c). The distal end of the vein was spatulated to increase the anastomotic surface between the 2 vessels. The artery was sectioned at its bifurcation with the posterior branch to obtain the widest possible arterial patch. The lengths and diameters of all vessels were measured.

#### 2.6.3. Transplantation

The external iliac arteries and veins were dissected and completely freed to obtain a length of 5 cm. A second dose of 5000 units of heparin was injected intravenously. A Satinsky clamp was positioned on the iliac external vein, and a phlebotomy was performed using a scalpel and Potz scissors. The vessel was washed carefully with heparinized saline solution. Vascular anastomoses were initiated with the end-to-side anastomoses between the utero-ovarian vein and the external iliac vein, followed by the artery on the same side. A protected atraumatic bulldog clamp was placed on the utero-ovarian vein and uterine artery near the anastomoses, in order to avoid backflow. The other side was performed in the same manner. The anastomoses of the first 10 ewes were performed with interrupted sutures, and the following ones with two continuous sutures in Prolene 6/0. If the anastomoses were not immediately watertight, separate sutures were used. Hemostatic mesh (Surgicel^®^) was placed over the anastomotic sites. If the uterine artery did not pulsate evenly along its entire length, a dose of papaverine, a musculotropic spasmolytic vasodilator, was sprayed directly onto the arterial wall. The clamps on the utero-ovarian veins and uterine arteries were removed. In the absence of immediate surgical complications, the vaginal slice was then sutured to the vagina, and the uterus was attached to the round ligament and anterior peritoneum with 2/0 Vicryl sutures.

### 2.7. Postoperative Monitoring

The ewes of series 1 and 2 were euthanized 3 h after the reperfusion, while those of series 3 were awakened following closure of the different plans. Daily veterinary monitoring of pulse rate, diuresis, food, and water intake, in addition to standing ability, was undertaken. Enoxaparin 0.4 mL was subcutaneously administered 6 h after the end of the procedure for anticoagulation, initially once a day and then every 12 h from the 17th ewe. Antibiotic prophylaxis by Septotryl^®^ was performed daily. A blood sample was taken every morning at 8 a.m. from the jugular vein. The ewe was reoperated on 4 days after transplantation.

### 2.8. Criteria for Evaluating Success

The uterus and anastomoses were evaluated after reperfusion using clinical and ultrasound criteria. The different anastomoses were dissected. For each artery, we retained a composite criterion of success including the tightness of the anastomosis, the pulsatility, the presence of a Doppler flow, and the absence of transfixing points or immediate thrombosis. For each vein, the criteria for success were the tightness of the anastomosis and the absence of transfixing points or immediate thrombosis. If any of the 5 criteria were not met, the anastomosis was considered a failure. We observed the recoloration of both uterine horns, which had to be pink or red to be considered successful. The graft was considered a success if the uterus recolored well, and if all anastomoses were successful.

### 2.9. Collected Data

We collected data on 40 autotransplants, from 6 November 2017 to 7 February 2022 (i.e., 151 anastomoses: 76 venous and 75 arterial). Age, weight, and the rank of the ewe were recorded. This last quantitative variable, reflecting the learning curve, was divided into two groups: the first 10 ewes versus the last 30, according to a receiver operating characteristic (ROC) curve. The different operative times were collected, and especially the time of dissection of the uterus, and the time of cold and warm ischemia. Next, for each vessel, we collected anatomic data, the caliber and length of the free pedicle, noted the primary operator who performed the anastomoses, the technique used (separate stitches or continuous suture), and finally whether catch stitches were required to make the anastomoses airtight. Complications that arose during anastomoses were collected, which were mainly tears and acute thromboses or spasms that required flushing of the vessel with heparin.

In addition, for the artery, we noted whether the dissection of the uterine artery was performed upstream of the bifurcation with the posterior branch of the internal iliac trunk, and whether papaverine was used. After removal of the vascular clamps placed at the time of the anastomoses, vasospasm of the artery was frequently observed. The return of normal arterial flow was observed either after a few minutes of waiting, or after the use of a papaverine spray on the arterial wall. Finally, the results of the Doppler ultrasound of the uterine arteries (positive or null diastole, reversed flow or no flow) were collected.

We analyzed these criteria for each ewe and for each vascular anastomosis, separating vein and artery, immediately after surgery, and then in the short term for those that were awake. Ewes that were euthanized because of major intraoperative complications, or having died from another cause, were excluded from the analysis.

A selection of different biological parameters was analyzed and included blood gases with lactates, hematology (hemoglobin (Hb), white blood cells (WBC), platelets (Pq)), and biochemistry (creatinine, urea, AST, blood glucose). Each sample was collected from the jugular vein at 7 different times: T0 before the intervention, T1 after the realization of the anastomoses, T2 6 h after the reperfusion, T3, T4, T5, and T6 at 8:00 a.m. the following days.

### 2.10. Statistical Analysis

Four analyses were performed: clinico-pathologic variables associated with successful arterial and venous anastomoses were determined intraoperatively and at 4th postoperative day, respectively. Categorical variables were expressed as *n* (%), and were compared by applying chi-square test (or Fisher’s test if the sample size was too small). Quantitative data were expressed as mean ± standard deviation (SE), and were compared using Student’s t test or a Wilcoxon test in case of non-parametric distribution. Parametric distribution was tested using the Kolmogorov–Smirnov test. Receiver operating characteristic (ROC) analyses of the significant quantitative factors were made to define threshold values. *p* values lower than 0.05 were considered as significant.

Significant variables that were identified by univariate analyses were entered into a multivariate logistic regression model in order to determine the variables that were independently associated with successful vascular anastomoses. All statistical analyses were carried out using XLStat Biomed software (AddInsoft V20.1, Paris, France)

## 3. Results

### 3.1. Immediate Success Criteria

#### 3.1.1. Arterial Anastomoses

Overall, 75 arterial anastomoses were performed: 59 were successful (78.7%), whereas 16 failed (21.3%). Most of the failures occurred in the first 10 ewes, where there were 43.8% of the failures and only 15.3% of the successes (*p* = 0.002) overall. Successful arterial anastomoses were significantly more frequent in young ewes (39.6 ± 6.5 vs. 45.4 ± 11.7 months, *p* = 0.01); however, looking specifically at our data, the first 10 ewes operated on were significantly older than the last (50 ± 12.94 months vs. 40 ± 5.39 months, *p* = 0.0315). They were also significantly more successful, with a shorter warm ischemia time (106.6 ± 30.2 vs. 135.3 ± 73.8 min, *p* = 0.002), when operated on by a vascular surgeon (71.2% vs. 43.8%, *p* = 0.04), when dissecting upstream of the posterior branch (81.4% vs. 56.3%, *p* = 0.008), and without any anastomotic complication (94.6% versus 72.7%, *p* = 0.04). The average time to perform the arterial anastomoses was 24 ± 13.0 min) in the success group, versus 30.2 min ± 14.7 in the failure group; however, this difference was not significant. The arterial diameter appeared slightly larger in the success group, although not significantly (5.3 mm ± 2.1 versus 4.7 mm ± 1.9, *p* = 0.32), and the pedicle length was approximately 8 cm. These results are presented in Table 1. Multivariate analysis failed to reveal independent factors that were associated with immediate successful arterial anastomoses.

#### 3.1.2. Venous Anastomoses

We achieved 82.9% successful venous anastomoses immediately after surgery. Similarly to arterial anastomoses, the majority of failures occurred in the first 10 ewes (*n* = 9, 69.2%). The ewes were significantly younger in the success group. The operating time (467.9 ± 80.8 vs. 525.8 ± 119.9 min, *p* < 0.03) and the warm ischemia time (104.2 ± 27.7 vs. 163.7 ± 76.5 min, *p* < 0.0001) were significantly shorter in the success group. Anastomoses times were twice as long in cases of failure (52.2 ± 34.3 vs. 26.0 ± 11.1, *p* < 0.0001). The performance of the anastomoses by a vascular surgeon, with continuous suture technique and no anastomotic complications, were also significantly more frequent in the success group. There was no significant difference in vessel length and diameter between the two groups. The free pedicle length was measured on average to be 7 cm, while the diameter was around 6 mm (Table 2).

Multivariate analysis identified the surgeon’s specialty (*p* = 0.04), intraoperative complications (*p* = 0.049), and warm ischemia length (*p* = 0.04) to be independent factors associated with immediately successful venous anastomoses. A warm ischemia period longer than 120 min and the presence of anastomotic complications significantly decreased the probability of venous anastomoses success, OR = 0.06 (95% CI = [0.003–0.99]) and OR = 0.05 (95% CI = [0.003–0.88]), respectively (Table 3).

### 3.2. Short-Term Success Criteria

#### 3.2.1. Arterial Anastomoses

At the 4th post-transplant day, 26 arterial anastomoses were analyzed. We recorded 65.4% (*n* = 17) of successful arterial anastomoses in the short-term analysis. There were eight failures related to thrombosis. In the failure group, the ewes had significantly lower body weights, longer operating times, longer cold ischemia times (49.6 min ± 7.6 vs. 43.1 min ± 7.7, *p*= 0.005), and warm ischemia (107.7 ± 28.2 vs. 90.3 ± 12.3, *p* = 0.04). Vessel size, remedial points, or the presence of reverse flow after reperfusion, were not influencing factors on the success rate. The results are presented in Appendix A. Multivariate analysis failed to reveal independent factors that were associated with successful arterial anastomoses at the fourth postoperative day.

#### 3.2.2. Venous Anastomoses

At the fourth post-transplant day, 25 venous anastomoses were analyzed. Twenty-one of the venous anastomoses were successful in the short-term analysis (84%). All failures were due to thromboses. The same parameters as for the arteries were significantly different between the two groups: shorter cold and warm ischemia times and the absence of any complication in the successful group. The results are presented in Appendix A. Multivariate analysis failed to reveal independent factors that were associated with successful venous anastomoses at the fourth postoperative day.

#### 3.2.3. Biologic Parameters

We collected data from 12 lactate samples at T1 (after reperfusion) and 11 at T2, 6 h after completion of the anastomoses. The crude levels are presented in Table 4. We then compared the average failure versus the average success according to the six sampling times. The maximum lactate level occurred at T2, i.e., 6 h after reperfusion. The results are presented in Figure 5 and in Appendix A. By comparing T2 to T1, ewes from the failed uterine autotransplants group had significantly increased lactate levels compared to ewes from the successful autotransplant group (3.58 ± 2.03 mmol/L versus 1.39 ± 0.8 mmol/L, respectively, *p* = 0.03). ROC analysis showed that an increase in lactate levels that was lower than 2.72 mmol/L predicted successful uterine autotransplant, with an 85.7% sensitivity (95% CI = [46.4–99.0]) and a 75.0% specificity (95% CI = [29.0–96.0]). There were no significant differences between both groups in regards to other biologic parameters.

### 3.3. Complications

Of our 40 autotransplants, two ewes presented complications of anesthesia: unexplained tachycardia and limb tremors. Both of these side effects were quickly resolved.

Twenty-four ewes were in the protocol with animal awakening (series 3). The ewes were monitored daily, and only 10 ewes (42%) survived to day 4. Fourteen ewes had complications and died of non-vascular causes, excluding them from analyses. Out of the fourteen failures, five were euthanized at the end of the procedure: three for uterine arterial problems (two accidental sectionings of uterine artery and one with no blood flow); one for an extensive small bowel wound; and one for an accidental ureteral section. Three hemorrhages occurred, resulting in one death at D0 and two at D1. One death occurred at D0 for extensive necrosis of the small intestine. Three deaths remained unexplained despite autopsies. The first revealed a white uterus, without hemorrhaging or organ necrosis. The second autopsy at D1 showed a white left horn, with a transfixing point at the level of the left uterine artery, and a leaky cervical anastomosis. The last one at D3 showed a uterus that was somewhat purple, with a transfixing point at the level of the right uterine artery, but without thrombosis. Finally, two deaths were related to uterine necrosis following thrombosis of the anastomoses (i.e., 16%). They occurred at D2 and D3, and the autopsy revealed three thromboses on the four vessels for each. These results are summarized in Figure 6.

## 4. Discussion

To our knowledge, this is the largest cohort of uterine autotransplantation used in a sheep model, which allowed us to analyze a large number of vascular anastomoses.

Our results showed that a warm ischemia time lower than 2 h, the absence of anastomotic complications, and the surgeons’ skills, were the main predictive factors for successful vascular anastomoses.

Uterine transplantation, as a surgical innovation, is subject to many challenges. Moore’s criteria [24] and IDEAL recommendations [25], including training in animal models, are mandatory before initiating any human transplant program. The excellent outcomes in uterine transplantations in the Swedish team, a leader in the field, are the result of long and meticulous preparation on animal models for more than 20 years [26]. The sheep animal model seems to be the most appropriate for surgical training on pelvic and uterine vessels of similar size to that of women [27].

In our model, the whole uterus was autotransplanted, and the anastomoses were performed mainly bilaterally, end to side, from the uterine artery or utero-ovarian vein, to the external iliac vessels. This procedure is similar to UTx in humans, except for the utero-ovarian vein, and is more comparable to the ovarian vein in terms of diameter and length. Immediately after surgery, we achieved a success rate of approximately 80% per vessel. This rate decreased in the short-term analysis, with 84% success for veins and 65% for arteries. Our success rate is consistent with the literature, which mentions 30% of failures resulting mainly from thromboses [7].

Although 40 cases were performed by our team on animals, this technique remains tricky; the slightest complication during venous anastomosis was predictive of failure. The longer the warm ischemia time was, the higher was the failure risk. This fact illustrates the correlation between the length of surgery and its complexity. The learning curve was faster for the vascular surgeon; vascular anastomoses were immediately successful from the 16th ewe following the time of his arrival, whereas nine venous anastomoses were necessary to obtain reproducibility with non-vascular surgeons. This had to be counterbalanced by the prior training from the team, which may have improved his learning curve. Nevertheless, training in microsurgery was required.

For the arteries, we did not find independent predictive factors of success for the anastomoses, either in the immediate postoperative analysis or in the short-term analysis. However, we found trends towards the same direction as for the veins: short operative and warm ischemia times, and realization of the anastomoses without any complication by a vascular surgeon. In contrast to veins, we had a high arterial anastomoses failure rate in our short-term analysis. Even if the arterial anastomoses seemed to be easier to perform than venous ones, nearly 40% of the arteries thrombosed within 4 days of grafting.

The learning curve of this procedure is important, and is represented by the rank of ewes. As we have already demonstrated with a ROC curve, a total of 10 ewe trials were needed to master the technique [17]. The rate of success increased from 12% to 87% for the veins, and from 15% to 83% for the arteries after the first 10 ewes. Some tips and tricks were identified progressively and consequently improved the prognosis of uterine transplantation. The use of bipolar coagulation was reduced; this subsequently avoided the use of pinching the vessels with our instruments during vessel dissection, reducing trauma and the risk of thrombosis. From the 11th case onward, we performed the dissection of the uterine artery upstream of the posterior branch of the iliac trunk, in order to obtain a larger arterial patch. A major step was the modification of operative technique by performing two continuous sutures instead of separate stitches. In case of uterine arterial spasms following the removal of the vascular clamps, papaverine spray was used from the 19th ewe onward. In order to decrease the risk of thrombosis without increasing the risk of hemorrhaging, the prophylactic anticoagulation protocol was adapted: first, the dose of heparin injected intravenously before clamping the uterine vessels was halved from ewe number seven onward, (10,000 to 5000 IU); secondly, a double preventive anticoagulation was implemented from the 17th ewe forward. We noticed cases of small intestine injuries, mainly due to the use of retractors. Careful exposure was performed after the death of the 19th ewe due to extensive necrosis of the intestine. Injuries to the bladder and ureter were also described, and led us to perform careful dissections of these organs. Although none of these were independent factors in the success of vascular anastomoses, they have allowed us to improve our surgical protocols.

For arterial and venous anastomoses in the immediate postoperative period, we found that the ewes were significantly younger in the success group than in the failure group. Arteriosclerosis or calcification of the vessel wall could be a reason for more failure in the older ewes. However, looking specifically at our data, we found that the first 10 ewes operated on were significantly older than the last, which could be a selection bias.

In our secondary analysis, we evaluated biological parameters. The differential of lactate levels above 2.72 mmol/L, between preoperative and 6 h after reperfusion, was predictive of failure. During reperfusion, lactates increase rapidly, and then return to values that are close to those measured before transplantation [23,28]. An excessive increase could be a poor prognostic factor for graft revascularization. This parameter could thus be used for postoperative management of the graft [29].

We obtained a high success rate for venous anastomoses, which validates the vascular feasibility. Unfortunately, we had many intra- and postoperative complications, and many ewes died of other or unknown causes. Although it usually takes 1 to 3 weeks for the patient’s general condition to stabilize after organ transplantation, we chose a time of only 4 days after surgery, considering that most complications occur in this period; this is a potential bias. As found in other studies, about half of the ewes survived the surgery; this could be explained by the length of the surgery, but also by the weakness of digestive organs in the sheep [30]. Few ewes were able to enter the short-term analysis, which prevented us from obtaining significant multivariate results.

Another limitation is that the success of uterus transplantation in this study was defined only by immediate vascular success. In uterus transplantation trials, success is defined by the delivery of a healthy child. The resumption of menstruation is also another sign of graft function recovery. Furthermore, secondary thromboses or hypoperfusion of vessels may occur more than 4 days after surgery.

The complexities of vascular anastomoses are obstacles for uterine transplantation to move from experimental surgery to current clinical practice. One of the trickiest parts of the procedure is the venous dissection. However, a single venous return may be sufficient [21]. Women’s uterus drainage is doubled via the uterine and ovarian veins (and not only via the utero-ovarian vein for sheep). Although most teams anastomose the uterine vein to the external iliac vein, using the ovarian vein may offer several advantages. This vessel is larger, around 4 mm in diameter [31], and flows through the upper pelvis; therefore, it is easier to access, and a longer pedicle is obtained by dissecting the vessel from the pelvis to the left renal vein or the inferior vena cava on the right. These factors could simplify and reduce the donor dissection time [32]. The contribution of each vessel to uterine outflow is poorly defined [31,33,34], but the report from Testa about a live birth that was achieved after using only the ovarian vein is promising [35]. Further testing is needed to confirm that ovarian veins are sufficient for uterine drainage, implantation, and normal pregnancy without complications. However, in order to use the ovarian vein of the donor, it would be necessary for her to be menopausal, or to remove her ovary at the same time. Yet, observational studies suggest that estrogen deprivation by bilateral oophorectomy may do more harm than good in post-menopausal and pre-menopausal women, as it carries an increased risk of cardiovascular morbidity and overall mortality [36].

With the aim of facilitating vascular anastomoses, aortic and cava patches have been attempted by some teams on sheep based on a deceased donor model, in order to increase the caliber of the anastomoses [18,37]. However, long pedicles are subject to a higher risk of twist and thrombosis.

As a perspective to improve this surgical procedure in a near future, robot-assisted laparoscopy may be considered. Indeed, this approach enhances precise and fine dissection deeper into the pelvis, and is particularly well-adapted for uterine veins dissection. Many teams are already using robots in UTx, especially for the most delicate part of the procedure, which is the removal of the graft from the living donor [15,38]. Robotic surgery has shown advantages in this application by improving dissection, surgeon comfort, and postoperative recovery [12,39,40]. For the surgery of the recipient, the complexity lies in the anastomoses of four vessels of small caliber. We speculate that this procedure could be facilitated by the robot, along with progress made in terms of the miniaturization of instruments. The setting up of a training platform that is specific to large animals for the simulation of intracorporeal surgery is still necessary. All of these improvements would make it possible to reduce the time of anastomosis, and thus limit the risks of warm ischemia to the survival of the uterus after transplantation [41].

## 5. Conclusions

The main factors for successful vascular anastomoses are a warm ischemia time of less than 2 h, and the absence of anastomotic complications. For this, it is essential that the anastomoses be performed by a vascular surgeon with previous training on animal models and in microsurgery. Lactate measurements during and after transplantation are a simple and rapid test that could be a predictor of graft revascularization. Long-term evaluation of the graft survival and gestation will be necessary to confirm our data.

## Figures and Tables

**Figure 1 jcm-11-05262-f001:**
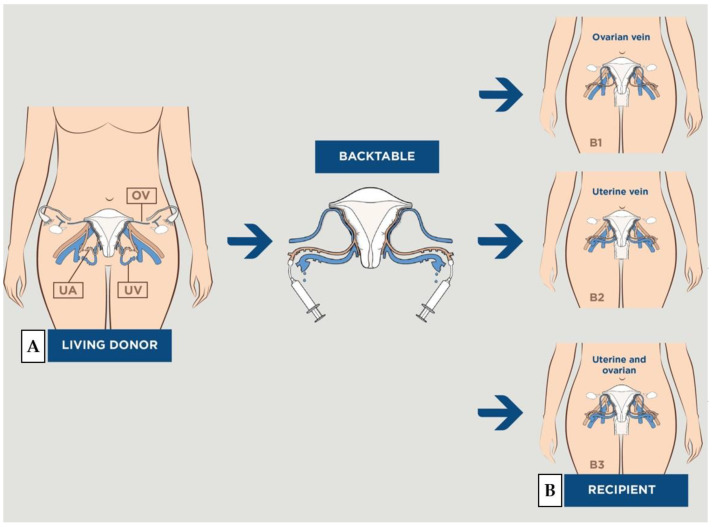
Schematic drawing of vascularization during uterus transplantation. EIA = external iliac artery; EIV = external iliac vein; I IA = internal iliac artery; OV = ovarian vein; UA = uterine artery; UV = uterine vein. (**A**) Vascularization obtained during uterine harvesting in the donor. (**B**) The uterus in place in the pelvis of the recipient with bilateral end-to-side anastomoses on the recipient’s external iliac vessels: (**B1**) with ovarian vein, (**B2**) with uterine vein, (**B3**) with both.

**Figure 2 jcm-11-05262-f002:**
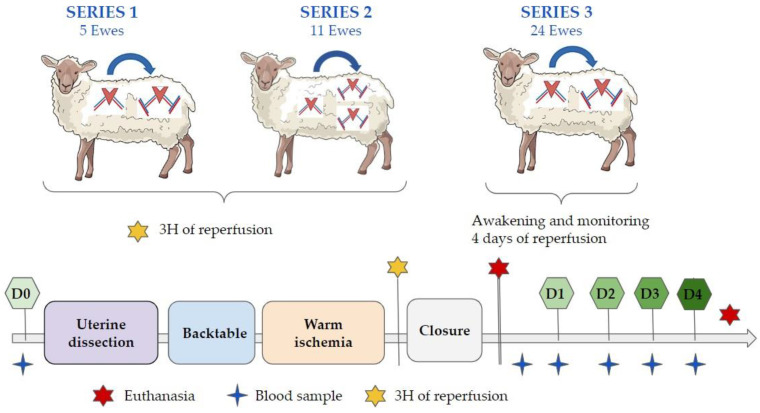
Experimental protocol and the main operating times. D0: day zero, D1: day one, D2: day two, D3: day three, D4: day four.

**Figure 3 jcm-11-05262-f003:**
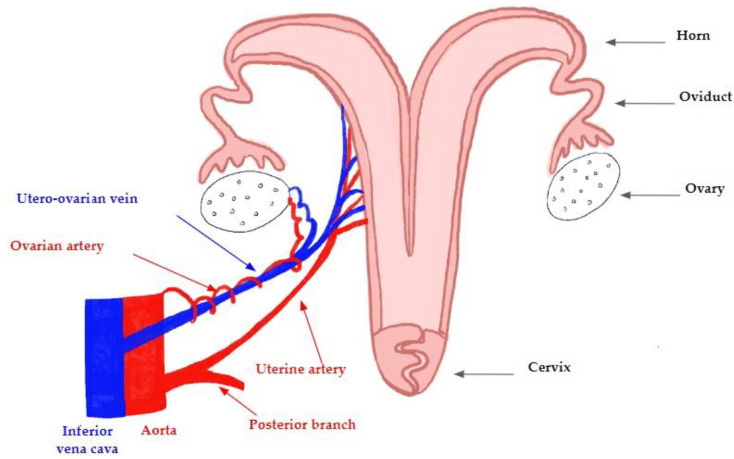
The anatomy and vascularization of the ovine uterus: the uterine artery and the utero-ovarian vein originate from the internal iliac vessels (trifurcation from the aorta and the vena cava, respectively).

**Figure 4 jcm-11-05262-f004:**
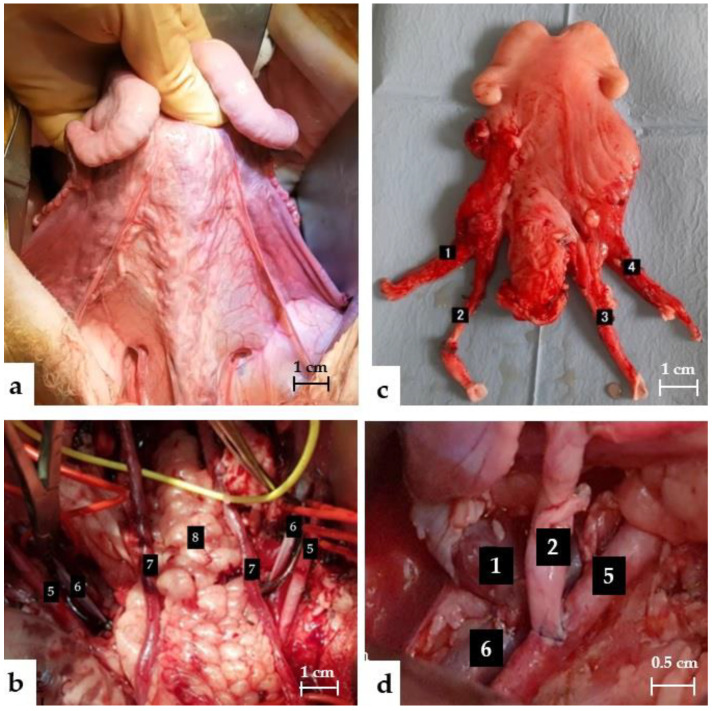
(**a**) Anterior view of the uterus before the dissection. (**b**) Pelvic view before anastomoses. (**c**) Posterior uterine view on the back table. (**d**) End of the anastomoses. 1 Left uterine vein, 2 left uterine artery, 3 right uterine artery, 4 left uterine vein, 5 external iliac artery, 6 external iliac vein, 7 ureter, 8 rectum.

**Figure 5 jcm-11-05262-f005:**
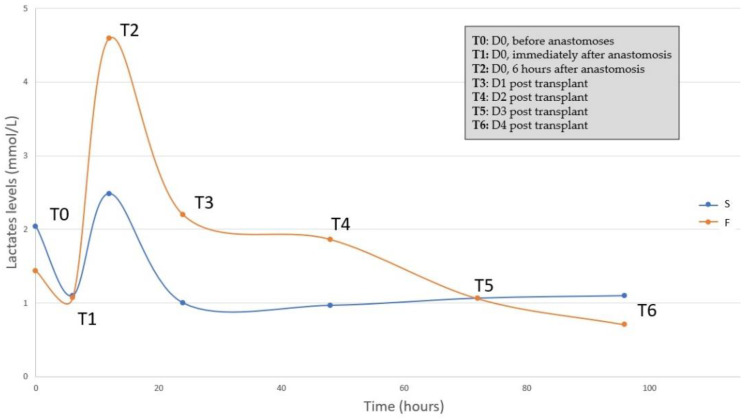
Lactate levels in success and failure autotransplant groups). S = success, F = failure.

**Figure 6 jcm-11-05262-f006:**
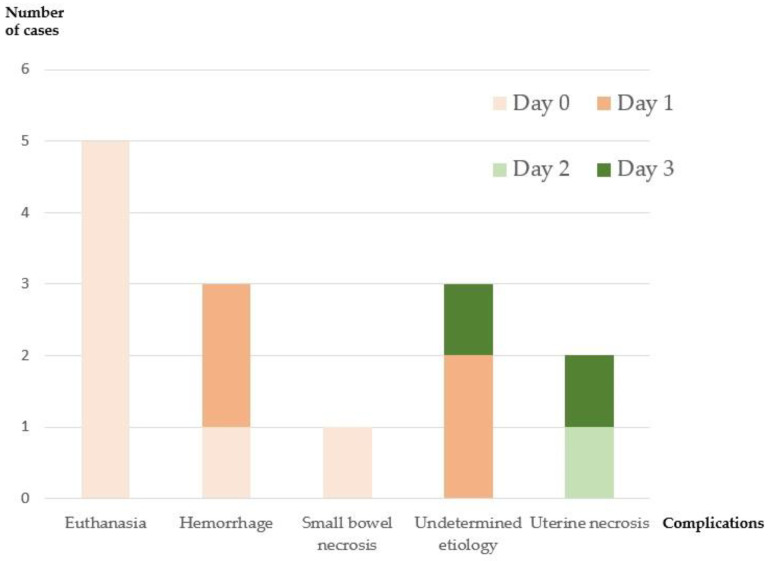
Description of the complications found in the 14 ewes of the third series. Each ewe had only one complication. Ewes that died of non-vascular causes were excluded from the short-term analyses (all except for those that died of uterine necrosis). Day 0 = day of the intervention, Day 1 = 1st postoperative day, Day 2 = 2nd postoperative day, Day 3 = 3rd postoperative day.

**Table 1 jcm-11-05262-t001:** Predictive factors for successful arterial anastomoses in the immediate postoperative period.

Variable	Success *n* = 59 (78.7%)		Failure *n* = 16 (21.3%)		*p*-Value
	*n* or Mean ± SD	Percentage or Range	*n* or Mean ± SD	Percentage or Range	
**Ewe characteristic**					
Age (month)	39.6 ± 6.5	29.1–70.0	45.4 ± 11.7	33.4–70.0	**0.01**
Weight	70.9 ± 11.6	52.0–97.0	71.8 ± 12.9	55.0–94.0	0.78
**Rank of ewes**					
1–10	9	15.3	7	43.8	**0.002**
>10	50	84.7	9	56.3	
**Anatomical data**					
Caliber (mm)	5.3 ± 2.1	1.0–10.0	4.7 ± 1.9	2.0–10.0	0.32
Length (mm)	83.4 ± 17.4	50–120	80.6 ± 24.5	45–150	0.61
**Transplantation**					
Operating time (min)	479.7 ± 86.5	353–678	472.9 ± 100.7	350–675	0.79
Time of uterine dissection	77.7 ± 16.6	50–118	74.0 ± 15.5	50–92	0.43
Cold ischemia	52.6 ± 31.3	20–150	59.7 ± 31.7	29–122	0.42
Warm ischemia	106.6 ± 30.2	75–197	135.3 ± 73.8	85–314	**0.02**
<120 min	51	86.4	8	50.0	
>120 min	8	13.6	8	50.0	**0.002**
**Anastomoses**					
Anastomoses time per vessel	24.0 ± 13.0	8–76	30.2 ± 14.7	12–66	0.1
Operator					
Non-vascular surgeon	17	28.8	9	56.3	**0.04**
Vascular surgeon	42	71.2	7	43.8	
Suture technique					
Separate points	8	13.6	5	31.3	0.1
Continuous suture	51	86.4	11	68.8	
Posterior branch					
Yes	48	81.4	9	56.3	**0.04**
No	11	18.6	7	43.8	
Papaverine					
Yes	6	10.2	3	21.4	0.25
No	53	89.8	11	78.6	
Missing			2		
Anastomoses complications *					
Yes	3	5.4	3	27.3	**0.02**
No	53	94.6	8	72.7	
Missing	3		5		
Remedial points					
Yes	15	26.8	4	33.3	0.65
No	41	73.2	8	66.7	
Missing	3		4		

* For each group: 2 tears and 1 thrombosis were resolved by heparin flush. *p*-Value < 0.05 is in bold.

**Table 2 jcm-11-05262-t002:** Predictive factors for successful venous anastomoses in the immediate postoperative period.

Variable	Success *n* = 63 (82.9%)		Failure *n* = 13 (17.1%)		*p*-Value
	*n* or Mean ± SD	Percentage or Range	*n* or Mean ± SD	Percentage or Range	
**Ewe characteristic**					
Age (month)	39.7 ± 6.6	29.1–70.0	49.4 ± 11.7	35.0–70.0	**<0.0001**
<40	43	68.3	3	23.1	
>40	20	31.7	10	76.9	**0.002**
Weight	70.4 ± 11.2	52.0–97.0	73.6 ± 14.2	57.0–94.0	0.37
Rank of ewes					
1–10	8	12.7	9	69.2	**<0.0001**
>10	55	87.3	4	30.8	
Caliber (mm)	5.93 ± 2.52	2.0–13.0	6.58 ± 2.84	3.0–13.0	0.42
Length (mm)	77.95 ± 22.4	35–130	68.33 ± 26.4	35–135	0.19
**Transplantation**					
Operating time (min)	467.9 ± 80.8	353–678	525.8 ± 119.9	350–678	**0.03**
Time of uterine dissection	76.5 ± 16.3	50–118	75.3 ± 18.7	52–118	0.82
Dissection time of the external iliac vessels	12.3 ± 7.3	2–36	14.9 ± 6.4	5–24	0.23
Cold ischemia	50.0 ± 28.8	20–150	67.0 ± 35.6	30–150	0.07
Warm ischemia	104.2 ± 27.7	75–197	163.7 ± 76.5	88–314	**<0.0001**
<120 min	55	87.3	4	30.8	
>120 min	8	12.7	9	69.2	**<0.0001**
**Anastomoses**					
Anastomoses time	26.0 ± 11.1	13–82	52.2 ± 34.3	20–129	**<0.0001**
<30 min	49	77.8	3	23.1	**0.0001**
>30 min	14	22.2	10	76.9	
Operator					
Non-vascular surgeon	15	23.8	11	84.6	
Vascular surgeon	48	76.2	2	15.4	**<0.0001**
Suture technique					
Separate points	2	3.2	3	23.1	**0.008**
Continuous suture	61	96.8	10	76.9	
Anastomoses complications *					
Yes	3	5.0	3	42.9	**0.001**
No	57	95.0	4	57.1	
Missing	3		6		
Remedial points					
Yes	11	18.3	2	28.6	0.52
No	49	81.7	5	71.4	
Missing	3		6		

* For the success group: 1 tear, 1 wound of a collateral branch, and 1 transfixing point removed. For the failure group: 2 tears and 1 transfixing point removed. *p*-Value < 0.05 is in bold.

**Table 3 jcm-11-05262-t003:** Multivariate analysis of factors predicting successful venous anastomosis immediately after surgery.

Variable	OR	IC95	*p*-Value
**Age of ewes (month)**			
<40	1		
>40	0.88	0.11–7.14	0.91
**Rank of ewes**	1		
1–10	1		
>10	4.17	0.14–100	0.41
**Operator**			
Non-vascular surgeons	1		
Vascular surgeon	29.3	1.17–731.91	**0.04**
**Warm ischemia**			
<120	1		
>120	0.05	0.003–0.88	**0.04**
**Time of dissection**			
<30 min	1		
>30 min	0.19	0.02–1.52	0.12
**Suture technique**			
Continuous suture	1		
Separate stitch	0.78	0.08–8.33	0.84
**Anastomotic complications**			
No	1		
Yes	0.06	0.003–0.99	**0.049**

*p*-Value < 0.05 in bold.

**Table 4 jcm-11-05262-t004:** Lactate levels at T1 (directly after anastomoses) and T2 (6 h after reperfusion). S = success, F = failure.

Rank of Ewes	Status	Lactates T1	Lactates T2	Delta
17	F	1.3	-	-
21	F	1.33	5.53	4.2
24	S	1.1	2.29	1.19
26	S	0.56	2.43	1.87
27	S	1.17	2.2	1.03
28	F	0.95	3.67	2.72
29	S	0.77	1.48	0.71
32	S	1.17	4	2.83
33	F	0.42	1.75	1.33
35	S	138	1.84	0.46
36	S	1.53	3.15	1.62
39	F	1.36	743	6.07

## Data Availability

The data presented in this study are available on request from the corresponding author.

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
