# Peer review of "Analysis of Predictive Factors for Successful Vascular Anastomoses in a Sheep Uterine Transplantation Model"

_jcm, 2022, doi:10.3390/jcm11185262_

Round 1

Reviewer 1 Report

The authors performed autologous uterine transplants in 40 ewes to investigate potential factors involved in the success of vascular anastomosis in uterine transplants. No independent immediate success factors were found for arterial anastomosis, and for venous anastomosis, immediate success factors included the surgeon's expertise and skills, intraoperative complications, and warm ischemic time. Short-term and long-term analysis of venous anastomosis could not be performed due to complications of about half of ewes.

Reviewers have some questions about this paper.

Although this analysis refers to the immediate success factors of vascular anastomosis, the definition of successful vascular anastomosis is ambiguous when the success of vascular anastomosis in uterine transplantation is defined as the resumption of menstruation, pregnancy, and labor. .. It is necessary to describe how to evaluate a suitable vascular suture for transplantation.

Factors involved in the immediate success of vascular anastomosis have been mentioned, but short-term and long-term factors have not been analyzed. What kind of research process is expected to be necessary in order to lead to transplantation?

The review states that thrombosis is one of the leading causes of post-uterine transplant complications. It also describes changes suggestive of uterine obstruction (changes in uterine coloration after vascular anastomosis) even in the absence of thrombosis. Can it be another cause of uterine thrombosis, such as an allergic reaction or vascular spasm?

Author Response

We would like to thank the reviewer for kindly accepting to review our revised manuscript. We also thank  for the important revisions they suggested. We feel that revisiting and reshaping the manuscript according to the reviewer’s recommendations has greatly improved its readability.

Point 1: The present analysis mentions immediate success factors for vascular anastomosis, but if successful vascular anastomosis in uterine transplantation is defined as resumption of menstruation, pregnancy, and delivery, the definition of successful vascular anastomosis is ambiguous. It needs to be stated how to assess appropriate vascular suture for transplantation.

Response 1:  We apologize for being unclear and confusing to the reader. We precised it Line 218-225: “For each artery, we retained a composite criterion of success including the tightness of the anastomoses, the pulsatility, the presence of a Doppler flow and the absence of transfixing points or immediate thrombosis. For the vein, the criteria for success were the tightness of the anastomoses and the absence of transfixing points or immediate thrombosis. If any of the 5 criteria were not met, the anastomosis was considered a failure. We observed the recoloration of both uterine horns which had to be pink or red to be considered successful. The graft was considered a success if the uterus recolored well and if all anastomoses were successful”.

To explain the limitations of our vascular success definition we added in discussion line 474-477 “Another limitation is that the success of the uterus transplantation in this study was defined only by immediate vascular success. In Uterus transplantation trials, Success is defined by delivery of a healthy child. Resumption of menstruations is also another sign of graft function recovery Furthermore, secondary thrombosis or hypoperfusion of vessels may occur more than 4 days after surgery.”.

Point 2: Factors involved in the immediate success of vascular anastomosis are mentioned, but shortand long-term factors are not analyzed. What future course of research is anticipated to be necessary to connect to transplantation?

Response 2: We have mostly obtained results on the immediate success factors of vascular anastomoses. As mentioned above, We added in discussion a sentence about this limitation.

We also added at the end of conlusion a sentence about future research: line 518-519 “Long term evaluation of the graft survival and gestation will be necessary to confirm our data.”

Point 3: The review states that one of the main causes of complications after uterine transplantation is thrombosis. It also describes changes suggestive of uterine inhibition (changes in the coloration of the uterus after vascular anastomosis) even in cases in which thrombosis is not present. Is there another possible cause of uterine thrombosis, such as allergic reaction or vasospasm?

Response 3: Thank you for this very pertinent question. To be clearer, we added Line 241-244: “After removal of the vascular clamps placed at the time of the anastomoses, vasospasm of the artery was frequently observed. The return of normal arterial flow was observed either after a few minutes of waiting or after the use of a papaverine spray on the arterial wall. We did not note any allergic reactions.”

Reviewer 2 Report

In this paper, Gal and colleges analyzed the data in their sheep uterine transplant model, and then proved the predictors of successful vascular anastomosis. This paper would be significantly strengthened by the maximum number of sheep uterine auto-transplant and its vascular anastomoses.

 However, it is well known that in most uterine transplant facilities, even when vascular anastomosis is performed by surgeons with excellent vascular anastomosis techniques, it rarely results in uterine graft loss, though each facility staffs doing their best for the surgeries. The result of this article, the shorter warm ischemia time improve the success ratio, may not have any great impact to move the uterine transplant field forward. And some ethical issues, results are not well written, thus those would be recommended to well revised.

 I have outlined additional opportunities to provide clarity and improve the manuscript below.

Specific comments

1)    Animal experiments should be conducted in accordance with the principles of the 3Rs (Replacement, Reduction and Refinement). Please provide the rationale for requiring 40 cases, especially 24 cases in the third series. (e.g., the number of significant differences was estimated in advance, etc.)

2)    Please state the reason for setting Day 4 as the date of euthanasia, because it usually takes 1-3 weeks for the patient's general condition to stabilize after organ transplantation. (e.g. all major complications after uterine transplantation occur within 4 days after surgery, etc.).

3)    The predictive factors for successful vascular anastomosis in sheep uterine transplantation are important for the aim of this study, but successful vascular anastomosis alone is not considered as successful at the initial stage of uterine transplantation, as the recipients can finally move to pregnancy and delivery after the successful long-term maintenance. For the future development of uterine transplantation, it would be important to discuss about the 14 cases (58%)  died of non-vascular causes in Series 3, if you could. It would be so helpful if you would give some comments about lab data, pathological findings (not only uterus, but also vessels, kidney, liver, pancreas, heart, heart, stomach, intestine, etc) in discussion. 

4)    Please also check if Cr, urea, AST, and Lactates are elevated in non-vascular death cases. If they are elevated or platelet counts are decreased, systemic organ damage due to DIC or other causes may have occurred.

5)    Please describe the improvement plan to reduce the number of deaths and to achieve long-term survival, if you could.

6)    Please revise Figure 1 until the differences can be understandable at a glance when printed on paper, as the details in Figure 1 are difficult to understand without zooming in,

7)    Are the scales and numbers in Figure 4 correct? the scale in 4d says the uterine vein diameter is 2.5 cm. Typo in 4c; 1,5 cm → 1.5 cm

8)    Are the numbers in 4d correct? The uterine vein is anastomosed to the external iliac artery.

9)    Line203  Please write the meaning of H6, N°17.

10)  Line221 ROC first written. Please write the meaning of ROC.

11)  It is understandable that plastic surgeons had poor results, but I wonder if the term "plastic surgeon" could be changed to "non-vascular surgeon" since it could be taken as an accusation or discrimination against plastic surgeons.

12)  Please consider to discuss about the reason why younger sheep have a higher success rate. (e.g. arteriosclerosis or calcification in the vessel wall, etc.)

13)  Please consider to add the Posterior branch in Figueure 3, as it is difficult to understand the Posterior branch, which is one of the points of this procedure.

Thank you for the opportunity to review.

Author Response

We would like to thank the Editor for kindly accepting to review our revised manuscript. We also thank the Editor and the reviewers for the important revisions they suggested. We feel that revisiting and reshaping the manuscript according to the reviewer’s recommendations has greatly improved its readability.

Point 1: Animal experiments should be conducted in accordance with the principles of the 3Rs (Replacement, Reduction and Refinement). Please provide the rationale for requiring 40 cases, especially 24 cases in the third series. (e.g., the number of significant differences was estimated in advance, etc.)

Response 1: Thank you for pointing out this element that must be justified. We used 3 differents series which have each complied with the 3Rs. The first one with 5 animals to establish a standardized operative protocol and confirm the feasibility of the surgery, the second one. The second series included 11 ewes and was specifically designed to compare the performance of one venous anastomosis versus two.  Our third series was designed specifically to compare 2 organ preservation fluids. However, this series was the first one with awake of the animal and we were thus confronted with postoperative complications. We precied it in the article Lines 115-121: “For the third series, the protocol was modified and the ewes were awakened at the end of the procedure. In this series, in accordance with the 3R (Replacement, Reduction and Refinement) principles, only 10 ewes were initially required to compare 2 organ preservation fluids They were euthanized 4 days after surgery. Fourteen additional  ewes that were euthanized for intraoperative complications or that died before 4 days were excluded from the short-term analysis.”

The number of 40 was necessary to perform multivariate analysis. We precised it line 265: “ We have chosen to include 3 series with a total of 40 ewes to be able to do multivariate analysis. ”

Point 2: Please state the reason for setting Day 4 as the date of euthanasia, because it usually takes 1-3 weeks for the patient's general condition to stabilize after organ transplantation. (e.g. all major complications after uterine transplantation occur within 4 days after surgery, etc.).

Response 2: As the reviewer noted, we have chosen 4 days after surgery to perform euthanasia as the purpose of our third study was to evaluate markers of short-term ischemia and reperfusion. As you very well mentioned, most major complications after uterine transplantation occur within 4 days after surgery (bleeding complications, thrombosis, or surrounding organ failure). Nevertheless, This 4-day delay is a potential bias because the organ is not fully stabilized. We added a sentence Lines  447-451: “Unfortunately, we had many intra and postoperative complications and many ewes died of other or unknown causes. Although it usually takes 1 to 3 weeks for the recipient's general condition to stabilize after organ transplantation, we have chosen a time of only 4 days after surgery in our third series for euthanasia, considering that most complications occurred in this period.”

Point 3: The predictive factors for successful vascular anastomosis in sheep uterine transplantation are important for the aim of this study, but successful vascular anastomosis alone is not considered as successful at the initial stage of uterine transplantation, as the recipients can finally move to pregnancy and delivery after the successful long-term maintenance. For the future development of uterine transplantation, it would be important to discuss about the 14 cases (58%) died of non-vascular causes in Series 3, if you could. It would be so helpful if you would give some comments about lab data, pathological findings (not only uterus, but also vessels, kidney, liver, pancreas, heart, heart, stomach, intestine, etc) in discussion. 

Response 3: You are absolutely right and it is important to discuss our failures and high rate of post complications. We specifically looked at the 14 ewes that died before 4 days and were excluded from the analysis in series 3. We have added a paragraph to our discussion : Line 455-457: “We noticed a high rate of complications, especially in our third series with awake of the ewes: They were due to  dissection difficulties, hemorrhage, bowel injury or secondary thrombosis “. Lines 468-471: “The cases of small intestine injuries were mainly due to retractors. Careful exposure was performed after the death of ewe #19 due to extensive necrosis of the intestine. Injuries to the bladder and ureter were also described and led us to perform careful dissection of these organs.” We unfortunately don’t  have more pathological findings than those explained in our results.

We added line 474-278 to explain the aim of uterus transplantation: “Another limitation is that the success of the uterus transplantation in this study was defined only by immediate vascular success. In Uterus transplantation trials, Success is defined by delivery of a healthy child. Resumption of menstruations is also another sign of graft function recovery. Furthermore, secondary thrombosis or hypoperfusion of vessels may occur more than 4 days after surgery.”

Point 4: Please also check if Cr, urea, AST, and Lactates are elevated in non-vascular death cases. If they are elevated or platelet counts are decreased, systemic organ damage due to DIC or other causes may have occurred.

Response 4: This is a very good point and we have individually checked the different biological parameters in non-vascular death cases.  We precised it line 385-386: “The blood samples showed no abnormalities that would indicate systemic organ damage due to Disseminated intravascular coagulation or other causes. “.

Point 5: Please describe the improvement plan to reduce the number of deaths and to achieve long-term survival, if you could.

Response 5: Based on the reviewer's helpful advice, we have detailed the improvement plan to reduce death and achieve long-term survival.

  • Following our first bleeding death, we adapted our prophylactic anticoagulation protocol: "To decrease the risk of thrombosis without increasing the risk of hemorrhage, the prophylactic anticoagulation protocol was adapted: first the dose of heparin injected intravenously before clamping the uterine vessels was divided by 2 from ewe n°7, (10 000 to 5000 IU) and secondly a double preventive anticoagulation was implemented from ewe n°17. Lines 465-469
  • The third ewe in series 3 died rapidly from extensive small intestinal necrosis. Special vigilance was a implemented to the bowel throughout the surgery (placement of retractors, uterine dissection time). We have added a dedicated sentence in the disccusion: " The cases of small intestine injuries were due to retractors. Careful exposure was performed after the death of ewe #19 due to extensive necrosis of the intestine. Injuries to the bladder and ureter were also described and led us to perform careful dissection of these organs." Lines 469-472
  • We have also already mention some improvements performed lien 456-465: “We noticed a high rate of complications, especially in our third series with awake of the ewes: They were due to dissection difficulties, hemorrhage, bowel injury or secondary thrombosis. Some tips and tricks were identified progressively and improved the prognosis of uterine transplantation. The use of bipolar coagulation was reduced and avoided pinching the vessels with our instruments during vessel dissection to avoid trauma and reduce the risk of thrombosis. From the 11th cases, we performed the dissection of the uterine artery upstream of the posterior branch of the iliac trunk to obtain a larger arterial patch. A major step was the modification of operative technique by performing two continuous sutures instead of separate stitches. In case of uterine artery spasm following the removal of the vascular clamps, papaverine spray was used from the 19th

Point 6:  Please revise Figure 1 until the differences can be understandable at a glance when printed on paper, as the details in Figure 1 are difficult to understand without zooming in,

Response 6:  We have modified figure 1 in order to make the differences between the 3 possibilities of venous drainage more obvious.

Point 7: Are the scales and numbers in Figure 4 correct? the scale in 4d says the uterine vein diameter is 2.5 cm. Typo in 4c; 1,5 cm → 1.5 cm

Response 7: Thank you very much for your rigorous observation, it is actually a typo error in the scales of figure 4. We put the correct scales for the 4 figures as follows: Figure 4: a,b and c: 1,5 → 1 cm; 4d 2,5 cm → 0,5 cm. Line 182

Point 8: Are the numbers in 4d correct? The uterine vein is anastomosed to the external iliac artery.

Response 8: You are correct and this is an mistake in the numbers of Figure 4d. It is definitely the uterine artery that is anastomosing to the external iliac artery and not the vein. Thank you for your careful reading. Lines 182-183

Point 9: Line203  Please write the meaning of H6, N°17.

Answer 9: We apologize for being unclear and and have specified the term H6 and N°17. We have modified these two points for “Anticoagulation with Enoxaparin 0.4 mL subcutaneously was administered 6 hours after the end of the procedure, initially once a day and then every 12 hours from the 17th ewe”. Lines 209-210

Point 10: Line221 ROC first written. Please write the meaning of ROC.

Response 10: We apologize for not being precise and have given the meaning of ROC : Line 230-231

Point 11: It is understandable that plastic surgeons had poor results, but I wonder if the term "plastic surgeon" could be changed to "non-vascular surgeon" since it could be taken as an accusation or discrimination against plastic surgeons.

Response 11: We fully agree with the reviewer 2 and thank him for his vigilance against specialty discrimination. We have corrected this in the whole manuscript and tables.

Point 12: Please consider to discuss about the reason why younger sheep have a higher success rate. (e.g. arteriosclerosis or calcification in the vessel wall, etc.)

Response 12: We fully agree with the reviewer that this is an outcome that needs to be discussed. In our univariate analysis for arterial and venous anastomoses immediately postoperatively, we found that the ewes were significantly younger in the success group than in the failure group. However, when we look closely at our data, we notice that the first 10 ewes operated on were significantly older than the last ones (50 ±12.94 month vs. 40 ± 5.39 months, p = 0.0315). This result is corrected by the variable "rank of the ewe" and no longer appears significant in the multivariate analysis. We have thus added a paragraph to our results and discussion:

  • “Successful arterial anastomoses were significantly more frequent in young ewes (39.6 ± 6.5 vs. 45.4 ± 11.7 months, p = 0.01), but the first 10 ewes operated on were significantly older than the last (50 ± 12.94 months vs. 40 ± 5.39 months, p = 0.0315).” Lines 284-287
  • “For arterial and venous anastomoses in the immediate postoperative period, we found that the ewes were significantly younger in the success group than in the failure group. Arteriosclerosis or calcification of the vessel wall could be a reason for more failure in the older ones. However, looking specifically at our data, we find that the first 10 ewes operated on were significantly older than the last, which could be a selection bias.” Lines 448-452

Point 13: Please consider to add the Posterior branch in Figueure 3, as it is difficult to understand the Posterior branch, which is one of the points of this procedure

Response 13: We agree with the reviewer#2 that this information is essential. We have thus modified the drawing and made appear the posterior branch of the iliac trunk.

Round 2

Reviewer 2 Report

Thank the authors for the revises. Your revised article seems so much better than the first one. Now the posterior branch is clarified on Figure 3. Then, I would like to add one more comment about the figure 3. The others are typo and small things.

I have outlined additional comments written below.

Specific comments:

Line 193: Please consider change from "non-vascular surgeons” to “the other non-vascular surgeons" on line 193, meaning other than gynecologist-obstetricians, if acceptable.

Line 226-227: “15th, 16th to the 40th “ ; Please superscript or unify "th" throughout, including nx and #x.

Line 299: Figure 3

It would be so helpful if you would add the cut line on the blood vessels in Figure 3.

(e.g., single-dashed-line: cut line before 10th ewe, double-line: cut line after 11th ewe)

Line 312: (Orascopic HDL™ Endehavour™) or (Orascopic HDL™, Endehavour™)?

Line 343: Maybe typo, Figure 4d “0,5cm” “0.5”.

Line 498:  at 4th postoperative day. at the 4th postoperative day.

Line 1006: ewe n7 → 7th ewe,

Line 1007-8: ewe n17 → 17th ewe.

Line 1009: ewe #19 → 19th ewe.

Thank you for the opportunity to review again.
